# Initial Healing Effects of Platelet-Rich Plasma (PRP) Gel and Platelet-Rich Fibrin (PRF) in the Deep Corneal Wound in Rabbits

**DOI:** 10.3390/bioengineering9080405

**Published:** 2022-08-20

**Authors:** Seo-Young Choi, Soochong Kim, Kyung-Mee Park

**Affiliations:** 1Laboratory of Veterinary Ophthalmology, Department of Veterinary Medicine, Chungbuk National University, Cheongju 28644, Korea; 2Laboratory of Veterinary Pathology and Platelet Signaling, Department of Veterinary Medicine, Chungbuk National University, Cheongju 28644, Korea

**Keywords:** cornea, corneal deep wound, platelet, PRF, PRP, SIS

## Abstract

Platelet concentrates (PCs), including platelet-rich plasma (PRP) gel and platelet-rich fibrin (PRF), are autologous blood-derived biomaterials containing numerous growth factors. This study aimed to evaluate the initial healing effects of PRP gel and PRF on deep corneal wounds. Thirty-three eyes from New Zealand white rabbits were divided into four groups: group 1, lamellar keratectomy (LK); group 2, LK + commercial porcine small intestinal submucosal membrane (SIS); group 3, LK + SIS + PRP gel; and group 4, LK + SIS + PRF. Postoperative clinical and histological findings were observed for eight weeks. Group 1 showed no neovascularization during the observation period, and incompletely recovered with a thin cornea. Group 2 showed active healing through neovascularization, and a thick cornea was regenerated through the sufficient generation of myofibroblasts. Although group 3 showed a healing effect similar to that of group 2, angiogenesis and subsequent vessel regression were promoted, and corneal opacity improved more rapidly. In group 4, angiogenesis was promoted during initial healing; however, the incidence of complications, such as inflammation, was high, and myofibroblasts were hardly generated in the corneal stroma, which adversely affected remodeling. In conclusion, while PRP gel is a safe surgical material for promoting remodeling through vascular healing and myofibroblast production in deep corneal wounds, the use of PRF is not recommended.

## 1. Introduction

The cornea is a transparent layer of the eye that possesses refractive characteristics, while maintaining a physical barrier between the eye and the environment [1]. When severe corneal injury occurs, the loss of transparency and consequent visual disturbances can occur [2]. Several studies have demonstrated the cellular mechanism of corneal stromal wound healing, which is initiated by the activation of keratocytes to active phenotypes, fibroblasts, and myofibroblasts, and ends by disappearance [3,4,5]. These processes are mediated by cytokines and growth factors, including TNF-α, IL-1, and TGF-β. Myofibroblasts play an important role in both beneficial and detrimental corneal wound healing [6]. They contribute to restoring the integrity of the cornea; however, they can also induce persistent scars. Thus, proper differentiation and the disappearance of myofibroblasts are critical for remodeling following severe corneal injury.

The cornea maintains avascular properties under physiological conditions [7]. However, when severe injury occurs, the balance between angiogenic and antiangiogenic factors that preserve corneal transparency is threatened, and neovascularization occurs as a healing process [8]. Generally, corneal neovascularization is an unacceptable condition to maintain transparency [9]. However, if serious damage occurs to the cornea, angiogenesis can provide beneficial effects for wound healing by providing oxygen and nutrients, which are essential for cellular proliferation, migration, and metabolic activities [10].

Platelets are well-known for their therapeutic properties [11]. The alpha granules in platelets release growth factors, including TGF-β, VEGF, EGF, and PDGF-AB, which initiate and modulate the wound-healing process [12]. These factors in platelets, especially TGF-β, regulate cell proliferation, synthesis of the extracellular matrix, angiogenesis, immune response, and disappearance [13]. Furthermore, TGF-β induces keratocytes to differentiate into myofibroblasts, which is essential for the severe stromal wound healing process [14]. Platelets have been clinically applied in the form of platelet concentrates (PCs), including platelet-rich plasma (PRP) and platelet-rich fibrin (PRF) [15]. PRP is an autologous platelet suspension obtained via two centrifugation steps. It can be applied in a liquid or solid form after an additional activating step [16,17,18]. In previous reports, PRP eye drops showed beneficial effects on ocular surface diseases by boosting corneal re-epithelialization [9]. In addition, there are a few case reports in which PRP gel forms are surgically applied to deep corneal ulcers in combination with biomaterials, such as amniotic membranes or bovine pericardium [19,20].

Similarly, as a second-generation PC, PRF simplifies the preparation process compared to PRP. PRF also contains growth factors and cytokines while PRF contains a high concentration of fibrin and leukocytes. PRF not only offers the beneficial clinical effects of PRP, but also fibrin scaffolds that serve as a supportive structure for tissue regeneration and the release of growth factors [21]. PRF can be obtained by a single centrifugation step, and anticoagulants are not required: the same is not true for PRP. Owing to these advantages, clinical applications of PRF have been reported in orthopedics and dentistry; however, the application of PRF in corneal healing is limited. PCs have beneficial effects in wound healing, oral and maxillofacial surgery, nerve regeneration, and orthopedic surgery [22,23,24].

Even if both PCs are similar, there is a difference in the components contained in PRF, such as platelets, leukocytes, and fibrin, compared with those in PRP. Therefore, PRP and PRF may have different therapeutic effects depending on the wound site. There are no reports on the effect of the use of PRP gel and PRF on stromal healing from deep corneal wounds; therefore, the two PCs were compared in this study. Moreover, the effects of PCs on myofibroblast differentiation and angiogenesis in deep corneal wound healing have not yet been reported. We hypothesized that growth factors and cytokines secreted by platelets would accelerate corneal wound healing. We aimed to evaluate the healing effect of two PCs, PRP clot and PRF, with the commercially available porcine small intestinal submucosal (SIS) membrane shield as a corneal stromal substitute, on deep corneal wound models and compare their clinical efficacy.

## 2. Materials and Methods

### 2.1. Animal Experiments

Thirty-three eyes from male New Zealand white rabbits, weighing 2.5–3 kg, were used. Rabbits were obtained from Doo-Yeol Biotech (Seoul, Korea) and reared at the Laboratory Animal Research Center of Chungbuk National University. Animals were kept under the following conditions: room temperature, 20 ± 2 °C; relative humidity, 50 ± 10%; air ventilation rate, 10 cycles/h; and 12 h light–dark cycle.

### 2.2. Study Design

The rabbits were randomly divided into four groups: group 1, lamellar keratectomy (LK); group 2, LK + commercial porcine small intestinal submucosal membrane (SIS) (Vetrix^®^ BioSIS, single layer discs, An-vision, Salt Lake City, UT, USA); group 3, LK + SIS + PRP gel; and group 4, LK + SIS + PRF. Group assignments were performed randomly. A schematic of this study is shown in Figure 1. Slit lamp biomicroscopic examinations were performed at 1, 2, 3, 4, 6, and 8 weeks after surgery, and histopathology was performed at 3 and 8 weeks after surgery. Fourier–domain optical coherence tomography (FD-OCT, RTVue, Optovue Inc., Fremont, CA, USA) images were obtained 8 weeks after surgery.

### 2.3. PRP Gel Preparation

A total of 8.2 mL of autologous blood was collected from the femoral vein of rabbits into two tubes. Eight milliliters of blood was put into a 15 mL conical tube containing 0.8 mL of 3.8% sodium citrate (Sigma, St. Louis, MO, USA) for preparation of PRP. Two hundred microliters of blood was collected into EDTA tubes for CBC examination (mean concentration of platelets in whole blood: 43.36 ± 4.63 × 10^4^/µL). The conical tube was then centrifuged twice using a tabletop centrifuge according to previously described protocols with some modifications [23,25]. Briefly, whole-blood centrifugation was performed at 1600 rpm for 12 min. After that, the supernatant above the RBC and buffy coat layer was separated, and the second centrifugation was performed at 2000 rpm for 10 min. After the second centrifugation, a platelet clot formed at the bottom of the bottle. The clot was then resuspended in 400 µL of supernatant platelet-poor plasma (PPP), and additional PPP was mixed according to the concentration of platelets. The final platelet concentration was adjusted at 100–150 × 10^4^/µL (mean concentration: 116.63 ± 4.48 × 10^4^/µL). Then, 400 µL of PRP was transferred into a 48-well tissue culture plate and 20 µL of 5% CaCl_2_ (Daejung, Siheung, Korea) was added to the dish to activate platelets. Finally, PRP gel was prepared after incubation at 37 °C, 5% CO_2_, and for 30 min in a 4-well plate.

### 2.4. PRF Preparation

Five milliliters of autologous blood was collected and inserted into a 6 mL red-top plain tube (BD, Franklin Lakes, NJ, USA). Blood samples were centrifuged immediately at 2700 rpm for 12 min, according to a previous report, with some modifications [26]. After centrifugation, fibrin clots formed above the RBC layer. The fibrin layer was separated from the RBC layer using forceps and transferred to a Petri dish. Then, the fibrous material was gently compressed in a Petri dish.

### 2.5. Surgical Procedures

All surgeries were performed under general anesthesia. Mask induction and maintenance of anesthesia were performed using isoflurane (Terrell™, Hana Pharm, Hwaseong, Korea). All groups had LK first. A 6 mm biopsy punch (Kai Medical, Dallas, TX, USA) was used to line the margin of the area for keratectomy. LK was performed manually with a 2.0 mm crescent knife (Diamatrix, The Woodlands, TX, USA). FD-OCT was utilized to confirm that the thickness of the remaining cornea was 100–150 µm (mean thickness: 120.44 ± 3.95 µm, wound about 2/3 thickness). In group 2, three layers of SIS membrane with an 8 mm diameter were secured on the LK site using 8-0 polyglactin 910 absorbable suture (Vicryl, Ethicon, Raritan, NJ, USA). Implantation of PRP gel and PRF was added to groups 3 and 4, respectively, before securing the SIS membrane. Systemic antibiotics (enrofloxacin, Baytril, Bayer, Leverkusen, Germany) and nonsteroidal anti-inflammatory drugs (Meloxicam, Metacam Inj, Boehringer Ingelheim, Ingelheim am Rhein, Germany) were administered for seven days. In addition, topical antibiotic eye drops (ofloxacin, Ocuflox, Samil, Seoul, Korea), non-steroidal anti-inflammatory drugs (diclofenac, optanac, Samil), and artificial tears (0.1% hyaluronic acid, Lacure, Samil) were administered for seven days after the operation.

### 2.6. Clinical Analysis of Cornea

Corneal vascularity (%) was evaluated using our criteria in the vicinity of the surgical site. Our corneal vascularization criterion score is 0: 0%, 1: 1–25%, 2: 26–50%, 3: 51–75%, 4: 76–99%, 5: 100%. The severity of the corneal opacity surrounding the surgical site was scored since it was impossible to analyze the corneal opacity score of the part covered by the SIS graft using the modified MacDonald–Shadduck scoring system with a score of 0 to 6 (Appendix A) [27].

### 2.7. Histopathologic Examination

The eye tissues were fixed with 4% paraformaldehyde. Paraffin-embedded samples were sectioned and stained with hematoxylin and eosin (H&E) using standard protocols. Immunohistochemistry was performed to detect alpha-smooth muscle actin (α-SMA; Cell Signaling Technology, Danvers, MA, USA), which demonstrates the presence of myofibroblasts. A peroxidase staining kit (VECTASTATIN^®^ Elite^®^ ABC Kit, VECTOR LABORATORIES, Burlingame, CA, USA) was used. The staining procedure was conducted according to the manufacturer’s protocol and included deparaffinization, antigen unmasking, H_2_O_2_ incubation, blocking non-specific binding, primary and secondary antibody incubation, ABC incubation, and counterstaining.

### 2.8. Statistical Analysis

Experimental data were analyzed statistically using analysis of variance (ANOVA) and GraphPad Prism 7 statistical program (San Diego, CA, USA). The statistical significance was set at a *p*-value = 0.05.

## 3. Results

### 3.1. Clinical Evaluation of Corneal Vascularization, Opacity, and Complications

Corneal neovascularization was observed in groups 2, 3, and 4 within two weeks; however, it was not observed in group 1 (Figure 2). Groups 3 and 4 showed faster angiogenesis than group 2 in the first week (Figure 3). Vascularization was the highest at two weeks in groups 3 and 4. However, in group 2, it was the highest at three weeks. In week 3, blood vessels covered most of the corneal surface in all groups, except group 1. However, in group 1, vascularization of the cornea hardly occurred during the entire experimental period. Four weeks after surgery, vascular retraction was pronounced, and the clarity of the cornea began to improve in groups 3 and 4. In contrast, in group 2, corneal vascularization peaked at three weeks and continued until four weeks. After six weeks, it decreased to a level similar to that of groups 3 and 4. In all groups, except group 1, vascularization was maintained at a similar level, approximately at a score of 1, from 6 weeks. These results suggest that PCs, including PRP gels and PRF, promote angiogenesis followed by rapid regression in the early healing stage of deep corneal wounds. It also showed that SIS induces vascularization, and the use of PCs further promotes vascularized healing of deep corneal defects. Without any surgical intervention using biomaterials, insufficient healing reactions appeared during all periods of the deep corneal wound in group 1.

One to two weeks after surgery, the corneal opacity score rapidly increased in all groups, except in group 1 (Figure 4). In particular, in the case of group 4, the opacity was higher than that of groups 2 and 3 due to corneal edema caused by intraocular inflammation at week 1. In groups 2, 3, and 4, as vascularization progressed, opacity increased significantly until the second week after surgery. In contrast, in group 1, local opacity was observed only at the surgical site. In the third week, the increase in corneal opacity was stagnant; however, in group 3, the transparency showed a tendency to decrease compared to the previous week. At four weeks, the corneal opacity in groups 2, 3, and 4 decreased. Among them, group 3 showed the fastest improvement, and group 4 showed the slowest recovery. The clarity of the cornea was dramatically improved from week 6, and at week 8, the opacity was similar to that at week 6 in all groups. In summary, group 3, using SIS with PRP gel, showed the most rapid recovery of corneal opacity from the third week after surgery. This was associated with the regression of the corneal blood vessels. When SIS was used alone, there was regression of blood vessels; however, recovery was delayed compared to that in group 3, in which SIS and PRP were used together.

Hypopyon was observed in some cases in groups 2, 3, and in particular, group 4, in the first two weeks, and was relieved after three weeks (Appendix A). More cases in group 4 showed stromal inflammation and ulcers in the third and fourth weeks of the wound healing process compared to groups 2 and 3 (Appendix A). In group 4, two eyes were excluded from the following observations due to corneal perforation in the third and fourth weeks (Appendix A).

### 3.2. Histopathological Analysis of Initial Corneal Wound Healing Process

H&E-stained sections of corneal tissues were evaluated for thickness, inflammation, and presence of myofibroblasts at both three and eight weeks after surgery. Three weeks after surgery, a thin layer of stroma, sparse arrangement of stromal cells, and absence of inflammatory cells were observed in group 1 (Figure 5). In groups 2 and 3, a relatively thickened stromal layer and diffuse distribution of a high density of myofibroblasts and inflammatory cells were observed. Stromal neovascularization was observed in groups 2, 3, and 4. In group 4, the stromal thickness was relatively thin compared to that in groups 2 and 3. Numerous inflammatory cells were found; however, fibroblastic cells were rarely observed. An increased number of neutrophils (heterophils) was distributed in groups 2, 3, and 4 compared to group 1 (Figure 6). In addition, group 3 had a lower number of neutrophils than groups 2 and 4, which, corroborating with the previous results, suggests that the remodeling of group 3 progressed faster than that of groups 2 and 4. Surgical intervention using SIS with or without PRP gel in groups 2 and 3 improved the ability to reconstruct a thick stromal layer by an active healing reaction at three weeks after surgery. However, there was no proper regeneration or insufficient healing responses in group 1. PRF in group 4 deteriorated corneal healing because it only induced severe inflammation and not regeneration.

Three weeks after surgery, immunohistochemistry was performed to detect myofibroblast differentiation. Figure 7 shows that α-SMA detection was observed only focally below the corneal basement membrane in group 1. In contrast, α-SMA was widely detected in groups 2 and 3 in the corneal stroma. α-SMA was rarely detected in group 4. Overall, SIS with or without PRP gel markedly accelerated myofibroblast differentiation of keratocytes for stromal cell restoration. In contrast, PRF treatment deteriorates myofibroblast differentiation and stromal regeneration responses in deep corneal wounds.

At week 8 after surgery, incomplete restoration of the thickness was observed in group 1. Inflammatory cells were not found, and sparse distribution of stromal cells were observed (Figure 8). In group 2, multicellular layer epithelial hyperplasia was observed. In addition, new stromal cell formation occurs below the epithelium. The presence of RBCs and surrounding endothelial cells indicates angiogenesis. A small number of neutrophils were present in group 2. Numerous stromal cells, lymphocytic-plasmacytic infiltrates, and macrophages were diffusely distributed in the corneal stroma. This suggests that chronic inflammation and the remodeling process had progressed. In group 3, the thickness of the LK area decreased compared to that in group 2. Angiogenesis was also observed, and inflammatory cells were rarely observed. The population of stromal cells was remarkably decreased. No epithelial hyperplasia was observed. Lymphocytic-plasmacytic infiltration and macrophages were reduced compared to those in group 2, and some of the cells mainly remained near the vessels in the stroma. These findings suggest that chronic inflammation reactions regress and remodeling responses proceed more rapidly than in group 2. In group 4, a few neutrophils and a diffuse distribution of stromal cells were observed in the stroma. Lymphocytic-plasmacytic infiltration and macrophages were diffusely distributed in the stroma. Epithelial regeneration was impaired in some areas, and basal cells are not well-observed. These findings in group 4 may have induced incomplete healing. In summary, group 3 showed the fastest healing process in terms of corneal reconstruction at eight weeks after surgery. Groups 2 and 3 exhibited active healing responses. This indicates that PRP gel accelerates the healing response in deep corneal wounds. In contrast, groups 1 and 4 demonstrated incomplete healing responses. This suggests that PRF adversely affects corneal regeneration and causes complications, such as inflammation.

### 3.3. FD-OCT Examination for Corneal Thickness

Irregular and thin corneal stroma was found in group 1, which suggests the inappropriate restoration of the cornea (Figure 9). The epithelial layer is also irregular and thicker than the normal corneal epithelium. Groups 2 and 3 showed relatively high corneal integrity, with enhanced stromal density and thickness. The central cornea remained thick as remodeling was still in progress. In group 4, irregular stromal density, abnormal stromal structure, and damaged Descemet’s membrane were observed. Thickened endothelium and fibrinous formations were also observed. This indicated improper restoration, decreased structural integrity, and signs of inflammation. The entire cornea and stroma were the thickest in group 2 and the thinnest in group 1 (Figure 10). In group 3, remodeling proceeded faster than that in group 2, which was not significantly different from the normal corneal thickness. In the case of group 4, the overall thickness of the cornea was greater than that of group 3; however, the thickness of the stroma was not significant due to the thickening of the endothelial layer. These results show that the corneal stroma cannot be fully restored in deep corneal wounds without surgical intervention. In addition, the use of SIS and PRP gel indicates that they are effective biomaterials that promote the restoration and remodeling of the cornea over several weeks. However, unlike PRP gel, PRF does not restore the corneal matrix and causes inflammation.

## 4. Discussion

Deep corneal wounds are serious conditions that can cause inflammation, infection, descemetocele, perforation, severe pain, and vision loss. Various surgical options have been studied owing to the need for prompt treatment of deep corneal wounds [28,29,30,31,32]. Keratoplasty using an allogeneic corneal graft is effective for the reconstruction of deep wounds; however, its supply is limited and there is a risk of immune rejection [33]. For these reasons, some biosynthetic materials, such as artificial corneas (keratoprostheses), ECM-based implants, fibrin, silk, and nanomaterials, have been studied [28,29,30,34,35,36,37]. Native biological matrices, including animal-derived corneas, amniotic membranes, SIS, urinary bladder mucosa, pericardium, and exogenous stem cells, have also been reported for corneal reconstruction [28,38,39,40,41,42,43]. However, there are still concerns about the poor effects on deep corneal wound healing and high-risk patients for rejection, or animal-derived cells and pathogens of xenogenic materials. Reconstruction of the cornea using the patient’s conjunctiva has also been reported [44]. It is strongly advantageous for infection and helps in the recovery of the stroma owing to the supply of blood vessels in the grafts [28]. However, visual outcomes and aesthetics are poor. Therefore, the demand for better materials for the recovery of deep corneal wounds continues to increase.

Platelets are known for their ability to release growth factors and proteins involved in wound healing. Numerous growth factors, including TGF-β, VEGF, and PDGF-AB, are released into the α-granules of platelets [23]. PRP gel promoted angiogenesis, and these factors are considered to be the reason for the rapid vascular healing in the PRP gel group in this study. Moreover, TGF-β is a key factor in corneal wound healing, especially in myofibroblast differentiation [14,45]. Thus, it is possible that TGF-β released by platelets contributes to the increased differentiation of myofibroblasts.

Alio et al. first introduced PRP eye drops for dry eyes [20]. Next, solid PRP was used for some cases of corneal disease in patients with corneal perforation [19,46]. Together with other biomaterials, such as amniotic membrane and Tutopatch, PRP has been suggested as a safe alternative that can be usefully applied in cases of severe corneal wounds and perforations. Similar to these outcomes, our results indicate that PRP has therapeutic effects in deep corneal wounds. Additionally, in this study, we confirmed that PRP gel in the corneal deep tissue induced rapid formation and retraction of neovascularization and accelerated stromal remodeling by inducing myoblast formation.

PRF is also a second-generation platelet concentrate discovered by Choukroun, which simplifies the preparation procedure [47]. Platelet activators, such as thrombin or CaCl_2_ are not required, in contrast to PRP gels. Previous reports have demonstrated that PRF membranes showed sustained release of growth factors, including TGF-β, PDGF, and VEGF, for seven days. Moreover, owing to their autologous characteristics and simple preparation process, PRF membranes have been used in oral, maxillofacial, and plastic surgery. Can et al. applied PRF to three human corneal descemetocele and ulcer cases, and the PRF membrane showed pain-relieving and anti-inflammatory effects [48]. After several months, corneal thickening was confirmed; however, two patients underwent penetrating keratoplasty for visual rehabilitation. In this study, compared to the control (group 1), PRF induced more neovascularization and thickened the cornea; however, it caused various undesirable outcomes. Unlike in PRP gel, the buffy coat layer was not removed from the PRF membrane. In addition, the concentrations of fibrin and leukocytes are high, and small amounts of RBC contamination can easily occur during PRF production. Furthermore, the platelet concentration in PRP can be controlled by counting and dilution during the preparation procedure. However, the one-step centrifugation and immediate solidification procedure for the preparation of PRF cannot regulate platelet concentration. It can be hypothesized that the high concentration of fibrin, undefined number of platelets, and a high number of leukocytes in PRF cause excessive inflammatory responses. In our final production step for the PRP gel, we used only 400 µL of plasma from 8 mL of blood. For the production of PRF, we used all (average of 2.6 mL) of the plasma from 5 mL of blood excluding the RBC portion (hematocrit), which showed an average of 49% of the total blood [49]. According to the previous report, the average concentration of fibrinogen, a precursor of fibrin in 3-month-old male New Zealand white rabbits, was approximately 1.54 ± 0.39 g/L [50]. Therefore, the amount of fibrinogen contained in the PRP gel and PRF used in our experiments can be estimated. Approximately 0.62 ± 0.16 mg (in 400 µL of plasma) and 4.00 ± 1.01 mg (in 2.6 mL of plasma) of fibrinogen should be present in the PRP gel and PRF used in this experiment, respectively. In conclusion, PRF has approximately 6.5 times more fibrin than PRP.

In this study, we used SIS with PCs to enhance their long-term maintenance. Alio et al. also used biomaterials to seal PRP clots at the wound site [19]. In our preliminary experiments, we found that grafting of PRP gel or PRF alone could not be maintained for long periods. When PCs alone were sutured on the cornea without using biomaterials, they degraded, disappeared, or were lost within a week. Therefore, this method is not suitable for deep corneal wound repair.

The SIS used in this study is a commercially available, bio-suitable, and biodegradable material, and is currently used as a treatment for deep corneas in veterinary medicine [40]. The SIS membrane contains various growth factors, including TGF-β, FGF-2, and VEGF [51,52]. In both groups 2 and 3, neovascularization and myofibroblastic differentiation were mostly observed in both groups. Thus, it may be possible to infer that various factors and physical features of the SIS membrane also contribute to enhanced clinical outcomes. In this study, when PRP gel was used together with SIS, the healing effects were synergistic. The rates of corneal vascularization, restoration of corneal transparency, and remodeling of the stroma were also included.

We focused on the early healing effects of PCs. Although myofibroblasts indicate repair phenotype, an excessive population of residual myofibroblasts can also cause corneal opacity through the secretion of irregular ECM proteins. After the proper restoration of the wound area, myofibroblast disappearance is required to clear the cornea. In this study, at week 8, corneal opacity in all groups had a score of approximately 1, and transparency was greatly improved. However, complete remodeling of the cornea takes several weeks to months. Therefore, further studies are needed to establish the long-term effects of SIS and PRP on corneal clarity and complete remodeling.

## 5. Conclusions

In conclusion, PRP gel combined with SIS membrane appears to have synergistic and beneficial effects on deep corneal wound healing. The SIS membrane alone also had advantageous effects; however, earlier neovascularization followed by improvement in corneal transparency was observed in the group treated with a combination of PRP gel and SIS. Its autologous properties, safety, low cost, and healing effects make PRP a promising therapeutic material. However, PRF membranes are not suitable for deep corneal wound healing owing to inflammatory effects.

## Figures and Tables

**Figure 1 bioengineering-09-00405-f001:**
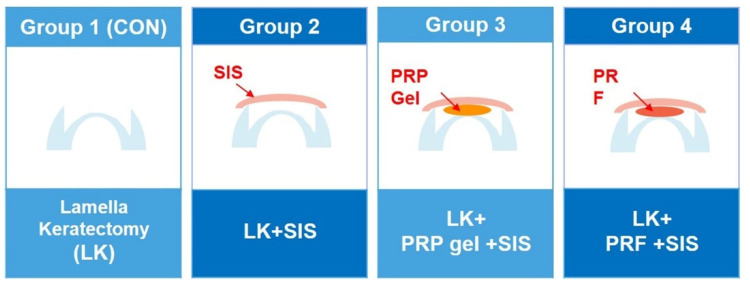
Schematic of the study. Group 1 had lamellar keratectomy only. SIS membrane grafting was used in group 2–4. PRP gel and PRF were applied in group 3 and group 4, respectively.

**Figure 2 bioengineering-09-00405-f002:**
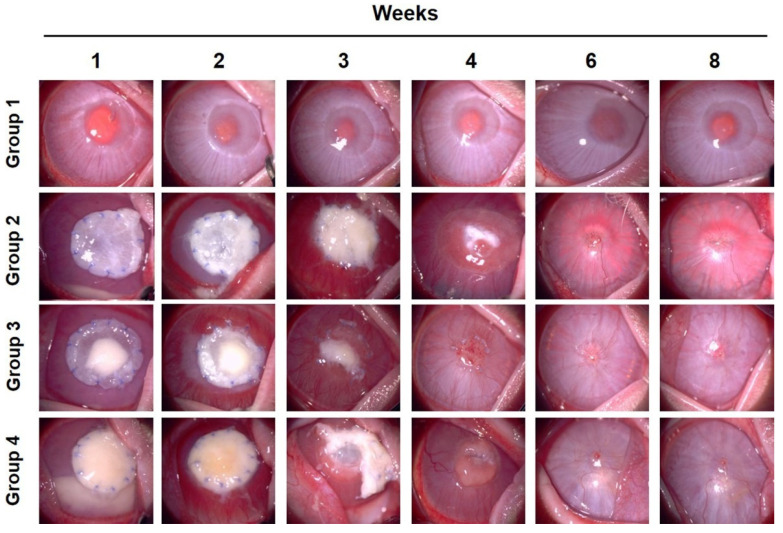
Clinical features of each group. Slit lamp bio-microscopic examination was done from 1 to 8 weeks after surgery. In group 1, angiogenesis was not induced; however, opacity was shown at the surgical site. In groups 2, 3, and 4, angiogenesis was actively induced up to the third week. In group 2, the opacity was noticeably improved from the sixth week, and in group 3, improvement in opacity was shown from week 4. However, in group 4, serious complications, such as descemetocele and inflammation, were seen from the third week.

**Figure 3 bioengineering-09-00405-f003:**
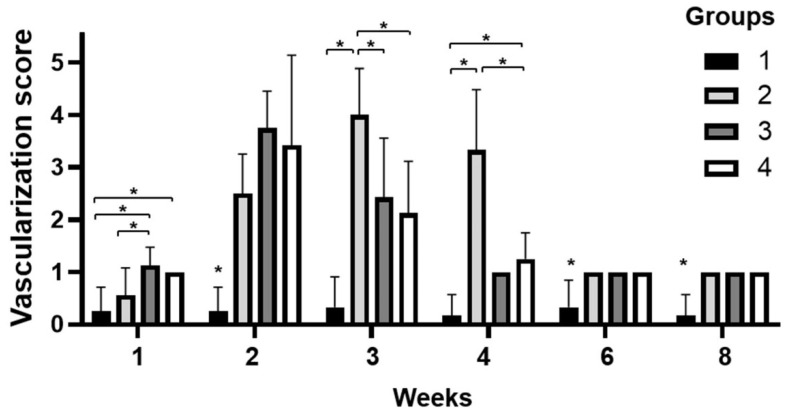
Corneal vascularization after surgery was scored. Neovascularization was established in groups 2, 3, and 4 within two weeks after surgery. Vascularization was generated faster in groups 3 and 4 than in group 2, and peaked at two weeks. In contrast, in group 2, vascularization peaked at three weeks. The rate of regression of vascularization was faster in groups 3 and 4 than in group 2. On the other hand, group 1 showed no neovascularization during all processes. Severity of opacity was increased in all groups. The number of samples in each group is indicated in Appendix A. *, *p* ≤ 0.05.

**Figure 4 bioengineering-09-00405-f004:**
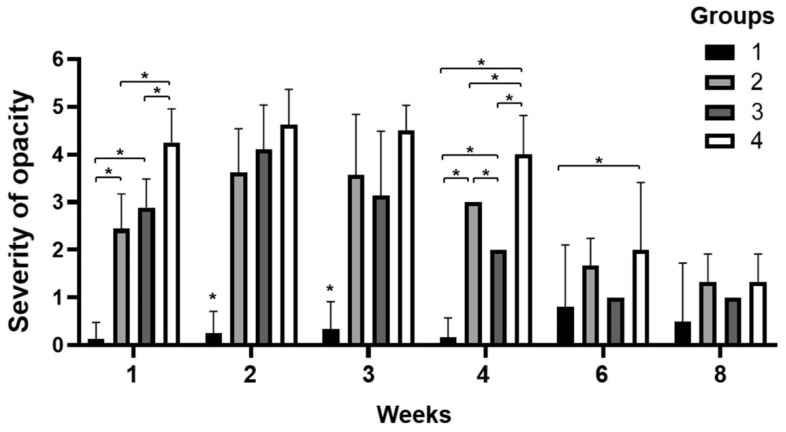
Severity of corneal opacity score was evaluated. In the first week, the opacity of group 4 was higher than that of the other groups. In the second week, the opacity of all groups except group 1 was increased. Although the opacity was decreased at 4 and 6 weeks, the opacity of group 4 was higher than that of the other groups. After six weeks, opacity was greatly reduced in all groups. The number of samples in each group is indicated in Appendix A *, *p* ≤ 0.05.

**Figure 5 bioengineering-09-00405-f005:**
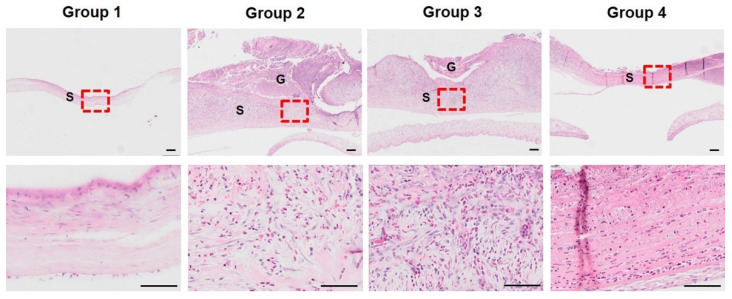
H&E staining of the cornea three weeks after surgery. No inflammatory cells were observed in group 1. Thickened stromal layer and diffuse existence of fibroblasts and neutrophils were presented in groups 2 and 3. The stroma thickness of group 4 is relatively thin compared to groups 2 and 3, and fibroblasts are rarely seen. The area marked with a red square in the 20× magnification was observed at 200× magnification. G = graft, S = stroma, Scale bar (black line) = 100 µm.

**Figure 6 bioengineering-09-00405-f006:**
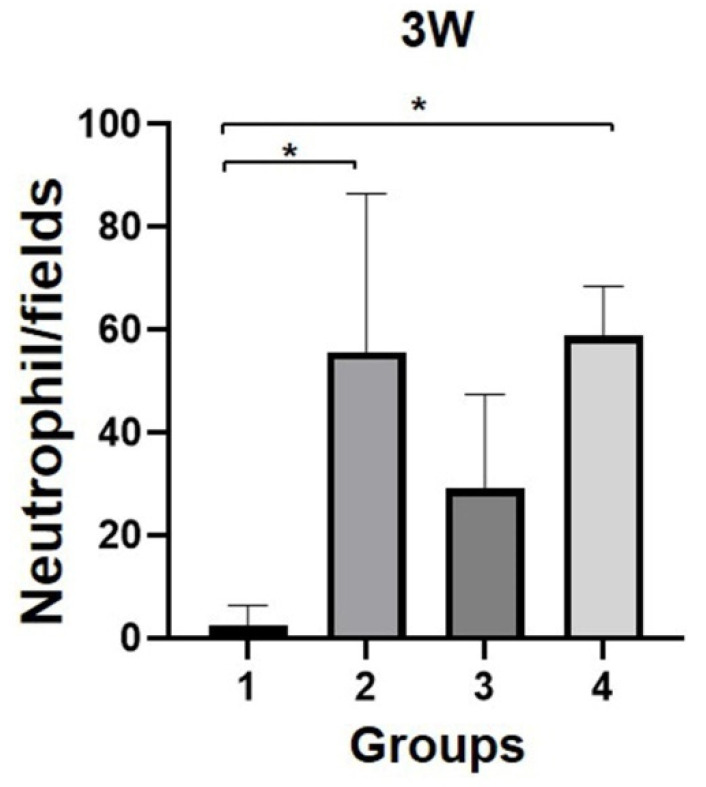
The neutrophils in the corneal stroma were counted at the high-power field. Neutrophils were increased in all groups except group 1, and group 3 showed a lower value than groups 2 and 4. n = 8 in groups 1, 2, and 4; and n = 9 in group 3. *, *p* ≤ 0.05.

**Figure 7 bioengineering-09-00405-f007:**
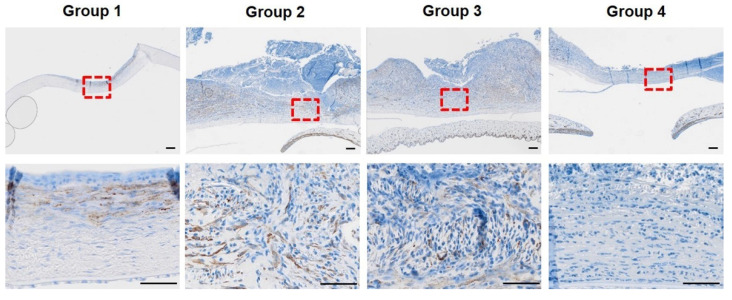
Immunohistochemistry of α-SMA of cornea as myofibroblast maker at three weeks after surgery was done. α -SMA positive cells were detected in brown. Few α-SMA was located focally below the basement membrane in group 1. α-SMA detection was remarkable in group 2 and 3. In group 4, α-SMA was hardly found. The area marked with a red square in the 20× magnification was observed at 200× magnification. Scale bar (black line) = 100 µm.

**Figure 8 bioengineering-09-00405-f008:**
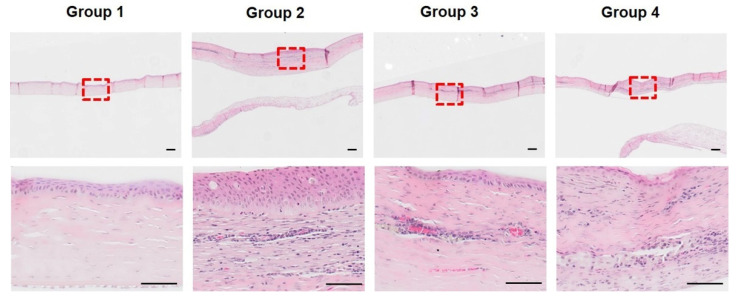
H&E staining of the cornea at eight weeks after surgery. Compared to other groups, group 1 had a thinner cornea and almost no inflammatory cells and stromal cells. Group 2 was the thickest and showed epithelial hyperplasia. High numbers of stromal cells remained angiogenesis within the stroma. In group 3, there was no epithelial overgrowth and angiogenesis; however, the number of stromal cells was greatly reduced and remodeling was completed compared to group 2. In group 4, the epithelium was not intact and neutrophils and stromal cells were present in the stoma. The area marked with a red square in the 20× magnification was observed at 200× magnification. Scale bar (black line) = 100 µm.

**Figure 9 bioengineering-09-00405-f009:**
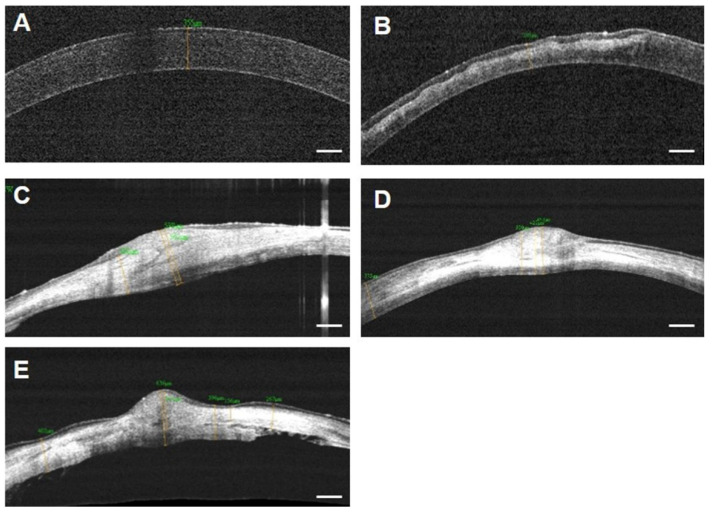
OCT images of all groups at eight weeks after surgery. (**A**) Normal cornea before surgery. (**B**) Group 1. Irregular epithelial and stroma layer indicates incomplete restoration. Full thickness of the central cornea is the least in all groups. (**C**) Group 2. Increased density of stroma is shown in the central area. (**D**) Group 3. Enhanced corneal integrity and intact epithelium are identified. (**E**) PRF group. Stromal surface is irregular, and the border of Descemet’s membrane is indefinite. Thickening of the corneal endothelial cell layer was observed, indicating that the endothelial cell layer was not properly remodeled due to inflammation. Scale bar = 250 µm.

**Figure 10 bioengineering-09-00405-f010:**
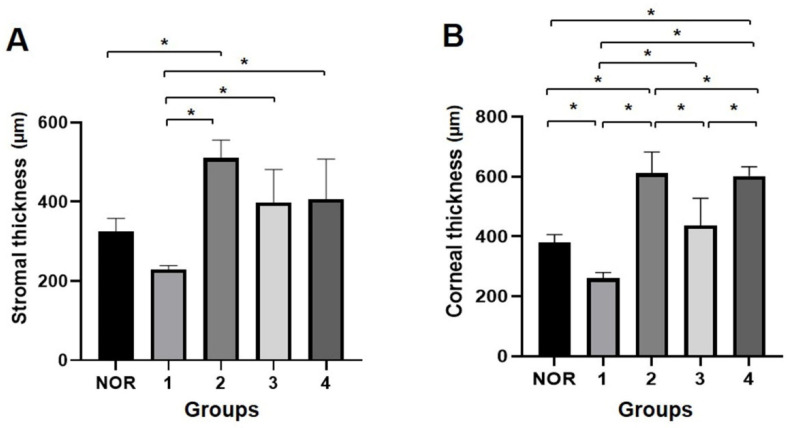
Thickness of total cornea and the stromal layer of each group were measured using OCT (**A**,**B**). The thickness of group 1 was thinner than that of normal cornea, and group 2 was the thickest. Groups 3 and 4 also showed a thicker cornea compared to group 1. n = 4 in NOR, group 1, 2, and 3; and n = 3 in group 4. *, *p* ≤ 0.05.

## Data Availability

Not applicable.

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
