# Peer review of "Initial Healing Effects of Platelet-Rich Plasma (PRP) Gel and Platelet-Rich Fibrin (PRF) in the Deep Corneal Wound in Rabbits"

_bioengineering, 2022, doi:10.3390/bioengineering9080405_

Round 1

Reviewer 1 Report

In this study, the authors evaluated PRP and PRF with SIS for corneal restoration in rabbits. The results indicated excess inflammation in both PRP and PRF, but they have therapeutic potential as a bioengineered material for deep corneal injury. The paper is pretty straight and simple. I do not find any significant problems with publication. I placed several questions/suggestions as minor points. Although it is not mandated, please follow/answer them upon the revision.

1) Generally, rabbits are maintained as an outbred colony. Different from mouse strain (identical), the rabbits are different. Did you use autologous PRP and PRF or from different rabbits? Please specify it.

2) Figure2. Did you show the same cornea from each group? From the images, 2w, 3w, 4w, and 6w in group 4 showed significant change. 

3) Line 187, "One to two weeks after surgery, the corneal opacity score rapidly increased in all groups, except in group 1 (Figure 4)." I thought the white one is sis graft. How can you examine the opacity with the sis graft? 

4) What are the a, b, c, d in figure3 and figure4? I cannot find the information.

5) Figure9 legend, Typo. (F) should be (E).

6) When you show the average, can you indicate sample number (N) in the legends?

7) Figure10. The author separately showed corneal thickness and stromal thickness. In normal corneas, it is not difficult. How can you measure them in deformed corneas such as in figure9?

Author Response

Response to Reviewer 1 Comments

In this study, the authors evaluated PRP and PRF with SIS for corneal restoration in rabbits. The results indicated excess inflammation in both PRP and PRF, but they have therapeutic potential as a bioengineered material for deep corneal injury. The paper is pretty straight and simple. I do not find any significant problems with publication. I placed several questions/suggestions as minor points. Although it is not mandated, please follow/answer them upon the revision.

1) Generally, rabbits are maintained as an outbred colony. Different from mouse strain (identical), the rabbits are different. Did you use autologous PRP and PRF or from different rabbits? Please specify it.

>> We thank the reviewer for reviewing our manuscript.

In this study, we used autologous PRP and PRF (not from different rabbits) to reduce errors in the results due to the immune response. We have added this information to the manuscript (page 3-4, line 104 and 122).

2) Figure2. Did you show the same cornea from each group? From the images, 2w, 3w, 4w, and 6w in group 4 showed significant change.

>> No. Previous versions have shown representative views rather than corneal photographs of the same rabbit. However, in accordance with the reviewer’s comments, we have replaced the corneas of the same rabbit with representative results (Fig. 2).

3) Line 187, "One to two weeks after surgery, the corneal opacity score rapidly increased in all groups, except in group 1 (Figure 4)." I thought the white one is sis graft. How can you examine the opacity with the sis graft?

>> Yes. The white one is the SIS graft. As it was impossible to analyze the corneal opacity score of the part covered by the SIS graft, we judged opacity from the surrounding part, except for the implanted part. This information has been included in the revised manuscript. (Page 4, line 149-152).

4) What are the a, b, c, d in figure3 and figure4? I cannot find the information.

>> Each of the different letters meant statistically significant however, these have been replaced with an asterisk (*) for clarity. (Figs 3 and 4)

5) Figure9 legend, Typo. (F) should be (E).

>> We thank the reviewer for pointing this out. This was a typographical error. We have changed the letter from (F) to (E).

6) When you show the average, can you indicate sample number (N) in the legends?

>> As per the reviewer’s comment, Table S.2 was added because the number of samples per week and group was too complicated to be indicated in the legends for Figs 3 and 4. For Figs 6 and 10, we have indicated the number of samples in the figure legend.

7) Figure10. The author separately showed corneal thickness and stromal thickness. In normal corneas, it is not difficult. How can you measure them in deformed corneas such as in figure9?

>> As the reviewer stated, it was difficult to measure corneal thickness with FD-OCT immediately after surgery. However, at the 8th week of surgery, corneal remodeling progressed to some extent and epithelialization was completed; therefore, it was possible to distinguish the epithelium from the epithelial layer by FD-OCT.

Reviewer 2 Report

General comments

            The authors use commercially available intestinal mucous membrane shields with platelet rich plasma (PRP) gel or platelet rich fibrin (PRF) to promote healing after deep lamellar corneal wounds. They find that PRP is more likely to promote healing of the corneal wound than PRF. PRF was associated with more complications such as descemetocele poor wound healing.

            The experiments are well performed and the data analysed appropriately.

            However, the rationale for the experiments is unclear and interpretation of differences in outcome is not provided.

            For instance, deep lamellar corneal wounds which involve removing a substantial amount of the corneal stroma without performing a graft are not wounds which are clinically relevant. Presumably, the PRP and PRF are being assessed as "corneal stromal substitutes" (ie artificial corneas) and these require mucous membrane shields to retain them in situ. However, this is not stated and therefore the aims of the study remain unclear.

It is also unclear why two different corneal substitutes are being compared

Specific comments

            Presumably PRP is a fibrin gel (ie a light fibrin clot) while PRF is a dense fibrin clot. Could the differences in wound healing simply be due to different concentrations of fibrin present in each model? Do the authors have data for the final concentrations of fibrin in PRF and PRP?  Fibrin is pro-inflammatory and a higher concentration would induce a stronger inflammatory response.

            The corneal opacity scoring is unclear and should be specified. The statistical evaluation using alphabetical lettering is unclear.

Author Response

Response to Reviewer 2 Comments

General comments

 The authors use commercially available intestinal mucous membrane shields with platelet rich plasma (PRP) gel or platelet rich fibrin (PRF) to promote healing after deep lamellar corneal wounds. They find that PRP is more likely to promote healing of the corneal wound than PRF. PRF was associated with more complications such as descemetocele poor wound healing.

The experiments are well performed and the data analyzed appropriately.

However, the rationale for the experiments is unclear and interpretation of differences in outcome is not provided. For instance, deep lamellar corneal wounds which involve removing a substantial amount of the corneal stroma without performing a graft are not wounds which are clinically relevant. Presumably, the PRP and PRF are being assessed as "corneal stromal substitutes" (ie artificial corneas) and these require mucous membrane shields to retain them in situ. However, this is not stated and therefore the aims of the study remain unclear.

>> We thank the reviewer for reviewing our manuscript. To clarify the purpose of our experiment, we have modified the Introduction section as follows (page 2, line 75-78).

“We aimed to evaluate the healing effect of two PCs, PRP clot and PRF, with commercially available porcine small intestinal submucosal (SIS) membrane shield as a corneal stromal substitute, on deep corneal wound models and to compare their clinical efficacy.”

It is also unclear why two different corneal substitutes are being compared

>> The following statements have been added to the text. (page 2, line 58-78)

>> Similarly, as a second-generation PC, PRF simplifies the preparation process compared to PRP. PRF also contains growth factors and cytokines. while PRF contains high concerntration of fibrin and leukocytes. PRF not only offers the beneficial clinical effects of PRP, but also fibrin scaffolds that serve as a supportive structure for tissue regeneration and the release of growth factors [21]. PRF can be obtained by a single centrifugation step, and anticoagulants are not required: the same is not true for PRP. Owing to these advantages, clinical applications of PRF have been reported in orthopedics and dentistry; however, the application of PRF in corneal healing is limited. PCs have beneficial effects in wound healing, oral and maxillofacial surgery, nerve regeneration, and orthopedic surgery [22-24].

Even if both PCs are similar, there is a difference in the components contained in PRF, such as platelets, leukocytes, and fibrin, compared with those in PRP. Therefore, PRP and PRF may have different therapeutic effects depending on the wound site. There are no reports on the effect of the use of PRP gel and PRF on stromal healing from deep corneal wounds; therefore, the two PCs were compared in this study. Moreover, the effects of PCs on myofibroblast formation and angiogenesis in deep corneal wound healing have not yet been reported. We hypothesized that growth factors and cytokines secreted by platelets would accelerate corneal wound healing. We aimed to evaluate the healing effect of two PCs, PRP clot and PRF, with commercially available porcine small intestinal submucosal (SIS) membrane shield as a corneal stromal substitute, on deep corneal wound models and compare their clinical efficacy.

Specific comments

Presumably PRP is a fibrin gel (ie a light fibrin clot) while PRF is a dense fibrin clot. Could the differences in wound healing simply be due to different concentrations of fibrin present in each model? Do the authors have data for the final concentrations of fibrin in PRF and PRP?  Fibrin is pro-inflammatory and a higher concentration would induce a stronger inflammatory response.

>> Unfortunately, we did not measure the fibrin content of PRP or PRF in this study.

 As mentioned by the reviewer, the negative effect of PRF on corneal healing may be due to the high concentration of fibrin, undefined number of platelets, and high number of leukocytes. Although PRF is simple to prepare, we think that these rough components contained in PRF are more likely to cause inflammation in wound healing compared to the pure PRP gel used in this study.

>> This has been added to the manuscript (page 12-13, line 392-399).

“ Unlike in PRP gel, the buffy coat layer was not removed from the PRF membrane. In addition, the concentrations of fibrin are high, and small amounts of RBC contamination can easily occur during PRF production. Furthermore, the platelet concentration in PRP can be controlled by counting and dilution during the preparation procedure. However, the one-step centrifugation and immediate solidification procedure for the preparation of PRF cannot regulate platelet concentration. It can be hypothesized that the high concentration of fibrin, undefined number of platelets, and high number of leukocytes in PRF cause excessive inflammatory responses. “

The corneal opacity scoring is unclear and should be specified.

>> According to the reviewer's comments, the corneal opacity measurement method has been listed in Table S.1.

The statistical evaluation using alphabetical lettering is unclear.

>> Each of the different letters meant statistically significant; however, these have been replaced with asterisks (*) for clarity (Figs 3 and 4)

Round 2

Reviewer 2 Report

The authors have clarified the rationale for performing these experiments. There are many assumptions made with regard to content of leukocytes and platelets in PRF but without repeating the entire work, these data can be determined in fresh samples. The authors could perform some further experiments to compare the cellular and fibrin content of PRP and PRF and add these data to the paper without repeating the in vivo work.  

Author Response

Response to the reviewer 2 comment

The authors have clarified the rationale for performing these experiments. There are many assumptions made with regard to content of leukocytes and platelets in PRF but without repeating the entire work, these data can be determined in fresh samples. The authors could perform some further experiments to compare the cellular and fibrin content of PRP and PRF and add these data to the paper without repeating the in vivo work.

>> We are thankful to the editor for this valuable comment.

>> Regarding the cellular contents of PRP and PRF, it is well known that PRF contains numerous leukocytes compare to pure-PRP because most leukocytes and RBCs are removed when preparing PRP. However, the concentration of fibrin (or fibrinogen) in the plasma should not change during PRP and PRF production because the plasma contents are maintained when preparing both PRP and PRF.

In our final production step for the PRP gel, we used only 400 ul of plasma from 8 ml of blood.  For the production of PRF, we used all (average of 2.6 ml) of the plasma from 5 ml of blood excluding the RBC portion (hematocrit), which showed an average of 49% of the total blood [1].

According to the previous report, the average concentration of fibrinogen, a precursor of fibrin in 3-month-old male NZW rabbits, was approximately 1.54±0.39 g/L [2]. Therefore, the amount of fibrinogen contained in the PRP gel and PRF used in our experiments can be estimated. Approximately 0.62±0.16 mg (in 400ul of plasma) and 4.00±1.01 mg (in 2.6ml of plasma) of fibrinogen should be present in the PRP gel and PRF used in this experiment, respectively. In conclusion, PRF has approximately 6.5 times more fibrin than PRP.

>> This description has been added to the manuscript (p.13, lines 405–413)

>> Unfortunately, contrary to what editors and reviewers asked for, we could not purchase and use a new laboratory rabbit for this revision. Instead, the estimated value was based on the previously published data because we had difficulties in purchasing new rabbits and conducting experiments within a short period.

1) We do not currently have the rabbits that were previously used in the experiments.

2) To measure the concentration of fibrin/fibrinogen in plasma from rabbits ourselves, we need to purchase new laboratory rabbits. To do so, we must obtain a new institutional IACUC permission, which takes at least 4–6 weeks. It is also not guaranteed that the IACUC will permit this additional experiment.

3) Please understand that it is impossible to perform the experiments within the given short period of 5 days. Fortunately, it was possible to estimate the difference between the fibrin concentrations of our PRP gel and PRF because the value of fibrinogen in the normal range was previously reported in rabbits.

References

  1. Özkan, C.; Kaya, A.; Akgül, Y. Normal values of haematological and some biochemical parameters in serum and urine of New Zealand White rabbits. World Rabbit Sci 2012, 20, pp.253-259.
  2. Petrova, Y.; Petrov, V.; Georgieva, T.M.; Ceciliani, F. Blood fibrinogen concentrations in New Zealand white rabbits during the first year of life. Bulg J Vet Med 2018, 21, pp.286–291
